# A retrospective two centre study of Birt-Hogg-Dubé syndrome reveals a pathogenic founder mutation in *FLCN* in the Swedish population

**Kristina Lagerstedt-Robinson**[1,2], **Izabella Baranowska Körberg**[3,4],
**Stefanos Tsiaprazis**[3,4], **Erik Björck**[1,2], **Emma Tham**[1,2], **Anna Poluha**[3,4], **Maritta Hellström Pigg**[3,4], **Ylva Paulsson-Karlsson**[3,4], **Magnus Nordenskjöld**[1,2], **Maria Johansson-Soller**[1,2], **Christos Aravidis**[3,4] *

1 Department of Molecular Medicine and Surgery, Karolinska Institutet, Stockholm, Sweden, 2 Department of Clinical Genetics, Karolinska University Hospital, Solna, Stockholm, Sweden, 3 Department of Immunology, Genetics and Pathology, Uppsala University, Science for Life Laboratory, Uppsala, Sweden, 4 Department of Clinical Genetics, Uppsala University Hospital, Uppsala, Sweden

* christos.aravidis@igp.uu.se

## Abstract

Birt-Hogg-Dube syndrome (BHDS) (MIM: 135150) is a rare autosomal dominant disorder with variable penetrance, caused by pathogenic variants in the *FLCN* gene. Only a few hundreds of families have so far been described in the literature. Patients with BHDS present with three distinct symptoms: fibrofolliculomas, pneumothorax due to lung cyst formation, and increased lifetime risk of kidney tumours. The aim of the current study was to estimate the incidence of BHDS in the Swedish population and further describe the clinical manifestations and their frequency. Splice variant c.779+1G>T was the most common pathogenic variant, found in 57% of the families, suggesting this may be a founder mutation in the Swedish population. This was further investigated using haplotype analysis in 50 families that shared a common haplotype. Moreover, according to gnomAD the carrier frequency of the c.779+1G>T variant has been estimated to be 1/3265 in the Swedish population, however our data suggest that the carrier frequency in the Swedish population may be significantly higher. These findings should raise awareness among physicians of different specialties to patients presenting with fibrofolliculomas, pneumothorax and/or kidney tumours. We also stress the importance of consensus recommendations regarding diagnosis and clinical management of this, not that uncommon, syndrome.

## Introduction

In late '70s three Canadian physicians introduced a novel genodermatosis syndrome characterized by the presence of multiple benign cutaneous neoplasms [1]. Birt-Hogg-Dubé syndrome (BHDS) (MIM# 135150), is an autosomal dominant disorder with unknown incidence. BHDS has as three main clinical features—the so called clinical triad; cutaneous hair follicle tumours, known as fibrofolliculomas or trichodiscomas, spontaneous recurrent pneumothorax in a pathological substrate of lung cysts and increased lifetime risk for kidney cancer [2–8].

**Data Availability Statement:** All relevant data are within the paper and its Supporting Information files.

**Funding:** The author(s) received no specific funding for this work.

**Competing interests:** The authors have declared that no competing interests exist.

Apart from the classic signs additional findings have been reported occasionally such as parotid oncocytoma, oral papules, skin melanoma, thyroid cancer and/or adenomas [2, 6, 7, 9–11]. The initial report in the literature regarding the association between BHDS and colon cancer with/or colon polyps is dated in mid '70s [12]. Since then, several cases of colon cancer and colon polyps in BHDS patients have been reported [3, 6, 13, 14], without strong evidence proving the association of this type of cancer with the syndrome due to conflicting outcomes.

The responsible genetic locus of BHDS was mapped with linkage analysis to chromosomal region 17p11.2 [13, 15], which subsequently led to identification of the causative *FLCN* gene [16]. The *FLCN* gene encodes the folliculin protein, a 64-kDa protein consisting of 579 amino acids that is expressed in a variety of tissues such as skin appendages, kidneys and lungs [16, 17]. Even though *FLCN*'s function is not fully understood, it has been suggested that *FLCN* acts as a tumour suppressor gene [18] interacting with FNIP1 and FNIP2 proteins, thus playing a key role in mammalian target of rapamycin (mTOR) signalling pathway [19–21].

The true incidence of the BHDS is still unknown due to clinical heterogeneity and variable penetrance, thus omitting a potential serious hereditary condition with significant pulmonary and kidney manifestations. Since its initial description approximately a few hundred families have been described. Reports of BHDS cohorts are valuable since they facilitate understanding of this condition, thus contributing to a better genotype-phenotype correlation. Herein, we present a large Swedish cohort of patients that were referred for genetic screening and consultation regarding BHDS.

## Material and methods

### Patients recruitment

The local Ethics Committee at Karolinska Institutet has approved this study (Dnr 2012/222-31/3), which followed the tenets of the Declaration of Helsinki.

Our cohort consists of 278 individuals from 78 seemingly unrelated kindreds that were referred to the Department of Clinical Genetics, Akademiska University Hospital, Uppsala and Department of Clinical Genetics, Karolinska University Hospital, Stockholm between years 2007–2019, for clinical and genetic investigation of BHDS, or carriership analysis of a known variant in *FLCN* gene in their family. All individuals included in the study reside from the same location, Stockholm-Uppsala region. All individuals included in this cohort study received genetic consultation from a clinical geneticist prior to genetic testing. A written informed consent was provided from each patient or minor, by their parents/legal guardians, for the study. In all index cases the reason for referral was strongly indicative for BHDS due to personal medical history and/or positive family history for at least one manifestation of the clinical triad of the syndrome. Of them, 125 are males and 153 females, with a mean age of 50 years that ranges from 15 to 97 years.

Referring clinics/departments were internal medicine, kidney/urologic clinics, pulmonary clinics and in some cases, patients were referred directly from primary health centres. After genetic verification of the diagnosis of the patient, further follow-ups were initiated by clinical departments regarding all aspects of importance for the disease. Accessible medical information regarding BHDS diagnosis (clinical examination data, imaging analysis and pathologic anatomical data) was reviewed individually for each patient.

### Genetic analysis

Genomic DNA was extracted from peripheral blood leukocytes using standard protocols. DNA samples were subjected either to direct analysis with Sanger sequencing or analysed using a multigene panel that included the *FLCN* gene. If no single or few base pair variant was

detected using sequencing analysis, screening for deletions/duplications was performed using multiplex ligation-dependent probe amplification (MLPA).

Regarding Sanger sequencing all coding exons of the *FLCN* gene were amplified using exon-specific primers (primers and PCR conditions are available upon request). Hence, non-coding regions, UTR's and enhancers were not sequenced nor evaluated. Direct Sanger sequencing was performed on both strands using Big-Dye terminator sequencing (v1.1, Applied Biosystems, CA, USA), and run on an ABI3130XL Genetic Analyzer (Applied Biosystems, CA, USA). The sequencing reactions were carried out according to the manufacturer's recommendations. Chromatograms were analysed using SeqScape v3.7 (Applied Biosystems, CA, USA) with the NCBI RefSeq NM_144997 as a reference sequence. Regarding analysis using a multigene panel the DNA sample was sequenced using whole genome sequencing (n = 7) using the Illumina HiSeq X Ten platform, using a 30× PCR-free paired-end WGS protocol. The patients were sequenced at Clinical Genomics, Stockholm, Sweden. Single nucleotide variants were called using Mutation Identification Pipeline (MIP) [22]. Three probands were analysed using a multigene panel at a commercial company (Invitae).

MLPA was performed with the available probe set for the *FLCN* gene (P256, MRC-Holland, Amsterdam, Netherlands) and carried out according to the provider's recommendations, with the exception that the PCR reactions were performed in a 25-μl reaction volume. Amplification products were quantified by capillary electrophoresis on an ABI3130XL Genetic Analyzer (Applied Biosystems, CA, USA) and the accompanying software. Tracing data were analyzed in GeneMarker software v1.7 (SoftGenetics LLC State College, PA, USA). The normalized quotients for the different probes were considered as a deletion when below 0.75 and indicative of duplication when above 1.3.

Regarding carriership analysis in genetically established familial cases, genomic DNA was examined by targeted mutation analysis using Sanger sequencing or MLPA.

## Haplotype analysis using microsatellites

Haplotype analysis was performed in a total of 50 individuals carrying the c.779+1G>T variant. Two first degree relatives carrying the variant were used to establish the familial haplotype (n = 28). If only one carrier in a family was available this single patient was screened (n = 22). Genomic DNA from the variant carriers were analysed using six polymorphic microsatellite markers surrounding the *FLCN* gene, located on chromosome 17p11.2 (D17S953, D17S1857, D17S740, D17S2196, D17S620 and D17S793). The markers were selected using the UCSC database, human assembly GRCh37. These markers span a total genomic region of 2,61 Mb where the FLCN c.779+1G>T (17,13Mb) variant is situated between markers D17S740 (16,99 Mb) and D17S2196 (17,26 Mb). Primers were pooled and amplified using Type-it Microsatellite PCR Kit according to the manufacturer's instructions (QIAGEN, Hilden, Germany). PCR-products were analysed using 3500xL Genetic Analyzer and GeneMapper v5 according to the manufacturer's protocol (Applied Biosystems, Thermo Fisher Scientific, Waltham, MA, USA). The variant carrying haplotypes were manually constructed and assessed between the different families/probands.

## Results

### Genetic analysis

In total 278 individuals were tested for pathogenic variants in the *FLCN* gene between 2007 and 2019. Of these 186 (67%) (Table 1, S1 Table) were found to have a pathogenic variant in the *FLCN* gene. The variant spectrum revealed eleven different pathogenic variants encompassing six of the 14 exons of the gene (Fig 1) where ten of the variants were identifiable using

**Table 1. Description of included cases in the study.**

| | |
|---|---|
| **Number of families (n)** | **78** |
| **Number of individuals (n)** | **278** |
| Men (n) | 125 |
| Women (n) | 153 |
| Age (y) | Mean 50, (15–97) |
| **Carriers (n)** | **186** |
| Men (n) | 95 |
| Women (n) | 91 |
| **Asymptomatic (n)** | **55 (30%)** |
| Age of asymptomatic carriers (y) | Mean 45, (15–89) |
| **Symptomatic (n)** | **131 (70%)** |
| Age of symptomatic carriers (y) | Mean 56 (20–97) |
| Fibrofolliculomas (n) | 88 (47%) |
| Pneumothorax (n) | 75 (40%) |
| Renal tumors (n) | 30 (16%) |
| Colon cancer and/or adenomas (n) | 9 (5%) |

sequencing analysis. Four separate families carried a deletion of the non-coding exon 1 that was detected using MLPA. The most common variant among families was the splicing variant c.779+1G>T that was found in 57% (44 of 78) of the families (Fig 2).

The, to our knowledge unrelated families, that carried the most common splicing variant were further analysed using haplotype analysis. Results from the haplotype analysis revealed that they shared one common haplotype surrounding the *FLCN* gene. The region spans between markers D17S740 to D17S620. This corresponds to a minimal common region of 0,58Mb to a maximal common region of 2,29Mb (Table 2).

## Clinical analysis

The clinical cohort includes healthy family members of BHD-families that were referred for carriership analysis. Of 278 individuals, 186 were carriers of a *FLCN* pathogenic variant and from the latter 55 exhibited no clinical manifestation as of July 2019. The age of asymptomatic carriers at the end of the study, ranges from 15 to 89 years old with a mean age of 45 years (Table 1).

On the other hand, symptomatic carriers constitute approximately 70% (131/186) of all carriers with a mean age of 56 years old and a range from 20 to 97 years. Regarding the clinical features of carriers; 47% (88/186) had cutaneous manifestations in form of fibrofolliculomas,

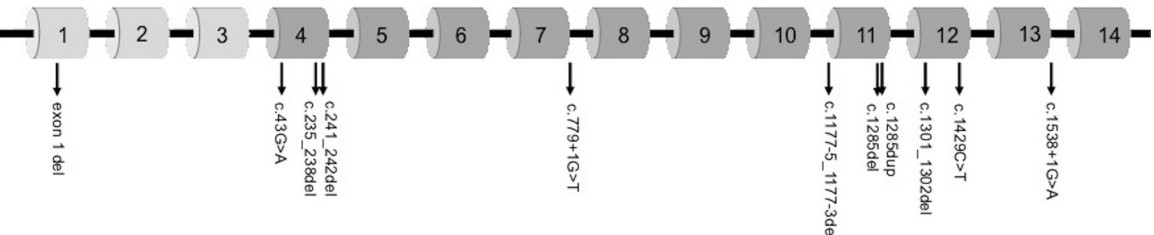

**Fig 1. Schematic figure illustrating the different variants in *FLCN*.**

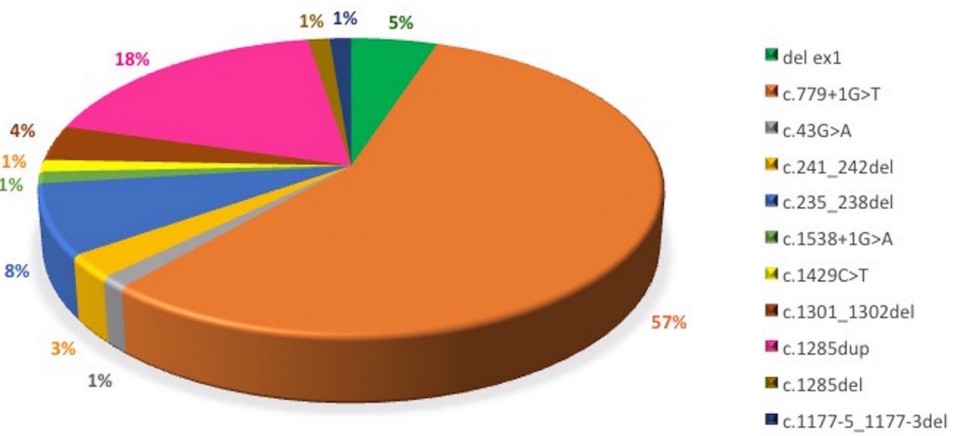

**Fig 2. Proportions of the different variant detected in our cohort.**

40% (75/186) had at least one episode of spontaneous pneumothorax and 26% (49/186) had pulmonary cysts in imaging control prior or after the genetic testing (Table 1, S1 Table).

Among all carriers 16% (30/186) had a medical record of a kidney tumour, that was confirmed by imaging techniques (CT, MRI, ultrasound) and/or histopathological examination. Renal tumours were identified before genetic analysis or after as a part of follow up controls. A histopathological report was available in 22 cases. In particular, eight cases had a hybrid type with elements of both oncocytic and chromophobe tumour, five patients had a chromophobe tumour, four patients had clear cell renal cancer. Additional histopathological findings in our cohort showed one case of papillary kidney cancer type 2, one case was hybrid oncocytoma/papillary tumour, one case of multifocal renal cancer could not be classified histopathologically, one case with bilateral benign renal tumour and there was one case of renal angiomyolipoma. The latter patient had a long diagnostic course since her initial differential diagnoses included Tuberous Sclerosis Complex (TSC) and lymphangioleiomyomatosis (LAM). The genetic analysis finally verified BHDS. As a part of the clinical investigation with imaging techniques, renal cysts were an additional finding in 9% (17/186) of carriers.

Other clinical diagnoses which were documented in carriers included thyroid cancer, parotid adenoma, chronic lymphoedema, liver haemangioma, skin melanoma, cutaneous fibrosarcoma, basal cell carcinoma of the skin that were present, one or more, in ten of 186 carriers. As a part of the investigation regarding association between colon cancer and BHD, we recorded nine of 186 (5%) carriers that had large intestine findings in the form of colorectal cancer and/or dysplastic adenomas (Table 1).

**Table 2. Haplotype analysis in families carrying the c.779+1G>T variant in the *FLCN* gene.** The shared haplotype is marked with italic and spans at least 0,58Mb up to 2,30Mb.

| Marker | Location chr:17 | | Haplotypes (bp) | | | |
|---|---|---|---|---|---|---|
| | (Mb) | 1 | 2 | 3 | 4 |
| D17S953 | 16,1 | 123 | 113 | 121 | 115 |
| D17S1857 | 16,42 | 177 | 177 | 180 | 180 |
| *D17S740* | *16,99* | *142* | *142* | *142* | *142* |
| *FLCN gene* | *17,12–17,14* | *c.779+1G>T* | *c.779+1G>T* | *c.779+1G>T* | *c.779+1G>T* |
| *D17S2196* | *17,26* | *160* | *160* | *160* | *160* |
| *D17S620* | *17,58* | *142* | *142* | *142* | *142* |
| D17S793 | 18,71 | 81 | 81 | 81 | 83 |

In all index cases there was a positive family history for at least one of three aspects of BHDS clinical triad (S2 Table). In 81% (63/78) of families there was a known history for skin lesions that resemble fibrofolliculomas, in 71% (55/78) a history of pneumothorax on one or more family members and 39% (30/78) of pedigrees had at least one individual who was affected with a kidney tumour. Regarding colorectal cancer/adenomas a positive family history was present in 14/78 (18%) families.

In our series of 44 families with the most common pathogenic variant, c.779+1G>T was found in 96 individuals. Of these 54% (52/96) had distinctive cutaneous findings of BHDS, 49% (47/96) had a history of at least one pneumothorax episode and 19% (18/96) were affected by renal tumours.

## Discussion

According to our knowledge the present study of 78 families is one of the largest cohorts ever published in current literature for BHDS [3–6, 10, 23–30]. In our series of 186 BHDS carriers we identified eleven different pathogenic and likely pathogenic variants that matched the clinical picture of carriers and their families (Table 1, S1 Table).

The most interesting finding in our study was the detection of a splice site variant c.779 +1G>T in 44 Swedish families (96 individuals). This particular pathogenic splice-site variant has been previously reported in four families [6, 11]. In the first report [6], describing three of the families, the variant was associated with a variety of cutaneous manifestations, renal tumours, lung cysts and spontaneous pneumothorax. The proband of the fourth family [11], had only cutaneous symptoms and further investigation by imaging techniques revealed cystic changes of the lungs and a single renal cyst. This patient did not have either a medical record of pneumothorax or family history for pneumothorax and/or renal cancer. In our cohort, 54% of the carriers of this specific splice site pathogenic variant exhibited cutaneous manifestations, 49% had a history of at least one pneumothorax episode by the time of genetic test and 19% were affected by kidney tumours. Among 44 families, 37 (84%) had members with a skin phenotype, 33 (75%) had members with a lung phenotype and in 17 families (39%) there was at least one member that was diagnosed with renal tumour. In seven of 44 families (16%) there was a history for colon cancer and/or polyps (S2 Table).

According to population databases (gnomAD) this variant has a carrier frequency of 5/ 64548 in Europe and of these 4 are of Swedish origin. This means that the carrier frequency of this variant in the Swedish population (1/3265) is significantly higher compared to the rest of the included populations (gnomAD) [31]. However, the frequency of this pathogenic variant in our cohort was strikingly high in this group of patients, which raised the suspicion that it might be a founder mutation. Haplotype analysis of the 44 families verified this hypothesis proving they are distant relatives, thus implicating that this variant is a founder mutation in the Swedish population. A speculation that the previously reported four families in the United States could be of Swedish descent is quite tempting.

The frameshift deletion of 4 base pairs, c.235_238del p.(Ser79Thrfs*50), has been described once in a large Finnish pedigree, in which numerous members were tested for the family variant [32]. Remarkably, all carriers showed lung manifestations only, while non-carriers were asymptomatic, illustrating the co-segregation of this variant with a pulmonary phenotype in the family. In our study, the variant was found in two different kindreds with a positive family history for the clinical triad of BHDS. However, whether these two apparently unrelated families have a Finnish origin or could be related to the previously reported pedigree is unclear.

Deletions of the first segment of the gene encompassing exon 1 of the gene have been previously reported in four families/index cases with BHDS [33–35]. Despite that, the three first

exons are non-coding, a deletion of the first exon eliminates the putative promotor of the gene, thus resulting in a substantial reduction of its expression [33]. Whole genome sequencing of one of our index cases showed that this deletion encompasses 8,1 kb ([GRCh37] del(17) (p11.2) chr17:g.17138220-17146328del). Phenotypic data in previous studies describe all three clinical hallmarks of the syndrome. In our series, two probands exhibited no significant symptoms but they had a positive family history for fibrofolliculomas, pneumothorax and kidney cancer. The other four had clinical symptoms consistent with BHDS.

An additional finding in our cohort was the detection of a 3 bp intronic deletion in position c.1177-5_1777-3del in one family. This variant has been reported before [28, 36] and is predicted to affect the splice acceptor site, thus leading to skipping of exon 11 and subsequently to a premature termination of folliculin synthesis. All three characteristic BHDS hallmarks have been described for that specific variant. In our family, the proband had only fibrofolliculomas, but there was a family history for skin lesions, spontaneous pneumothorax and lung cysts as well.

The most common duplication/deletion is in the "hot spot" c.1285 of the gene. This frequent variant, which has been observed in various cohorts with a variable frequency [6, 10, 14, 16, 27], was found in 15 unrelated pedigrees in our study. In particular, 14 families had the duplication of a cytosine at c.1285dup p.(His429Profs*27) and one family had the deletion of a cytosine at this position, c.1285del p.(His429Thrfs*39). The clinical spectral of analysed eight families included all typical signs for BHDS.

Other pathogenic variant that we detected in single families were a frameshift 2 bp deletion c.1301_1302del p.(Glu434Valfs*21) that has been reported once before [27] a nonsense variant c.1429C>T p.(Arg477*) (that has been detected previously [10]) and the three different novel variants c.241_242del p.(Met81Valfs*18) and c.1538+1G>A, c.43G>A p.(Gly15Ser).

Regarding the clinical description in general, distinctive cutaneous manifestations are the main features of the syndrome. These skin lesions are benign tumours of hair follicles named as fibrofolliculomas, trichodiscomas and acrochordons. They usually appear in the third or fourth decade of life as multiple white papules on the face, with a more common localization on the nose and cheeks, but they can also be visible on the neck, ears and upper trunk. They do not have any clinical implication and they are excised for psychological and cosmetic reasons with ambiguous outcomes [37]. In our series nearly half of all BHD carriers presented cutaneous lesions by the time of genetic investigation. Confirmation of these lesions performed either by clinical examination from a dermatologist and/or by histopathological examination.

Association between *FLCN* variants and pulmonary symptoms is well established, indicating that carriers have an up to 32-fold risk for spontaneous pneumothorax compared to general population [3], together with a high predisposition to develop lung cysts [3, 5, 6]. Especially, up to 89% of patients with BHDS can develop multiple pleural blebs and subpleural bullae localized mainly in basal segments of the lungs, that are considered precursor lesions to pneumothorax [6]. Radiographic investigation of these lesions regularly misleads physicians towards the diagnosis of emphysema. In our cohort 40% of carriers had a history of spontaneous pneumothorax. Moreover, an ascertainment with CT-thorax of BHDS carriers with or without a history of pneumothorax revealed that up to 26% of them had emphysematous-like bullae in their lungs. This outcome is lower than in previous reports [5, 6, 13, 23], nevertheless it should be mentioned that not all carriers were subjected to imaging analysis.

Several explications to the underlying molecular mechanism of cysts development in the lungs, have been suggested. FLCN protein (folliculin) has a tumour suppressor activity interfering with the mTOR signalling pathway, a fact that provides a plausible explanation for cyst formation [19]. Additionally, any variant which leads to a significant decrease of FLCN expression may induce the inflammatory process in alveolar walls or even lead to matrix degradation

and remodelling [32, 38]. Furthermore, a similar conclusion from another study suggests that pulmonary bullae imply a deficiency in the structural proteins of the lung wall, which may be elements of the cytoskeletal network [16]. Finally, an interesting explanation suggests that the folliculin may have a ciliary role, thus resulting in pulmonary cyst formation [39].

The most important life-threatening consequence of the syndrome is a significantly increased lifetime risk up to 34% for kidney cancer or a 7-fold higher risk compared to normal population [3, 4, 6, 13]. The tumours described in literature are multifocal, bilateral, with a slow progress and low metastatic potentional [37]. Median age of diagnosis is 48 years with a range from 31 to 71 years [10]. The earliest reported case for BHDS is a young patient that was diagnosed with renal cancer at the age of 20 years old [13]. Genetic screening where renal tumour was the only clinical symptom in the index patient revealed a *FLCN* variant in eleven of 21 cases, emphasizing that BHDS is common in this disease. In the current study, carriers were diagnosed with renal tumour in 16% of cases and the youngest affected individual was 20 years old. Overall incidence of renal tumours in our study corresponds to previous reports, however more cases need to be documented in order to have a more accurate estimation of the occurrence of kidney tumours in BHDS.

According to our knowledge, only few reports for renal cysts in BHDS suggest that up to 45% of carriers may have these usually asymptomatic lesions [7]. In our cohort the frequency of renal cysts in carriers were 9%, but the number of examined radiologically individuals is small and not all carriers performed radiographic kidney investigation during the study period.

Even though the underlying mechanism for renal carcinogenesis is still unclear, several researchers suggest that folliculin's role in mTOR pathway is related to kidney tumorigenesis by inactivating variants in *FLCN* gene coupled to a secondary loss of heterozygosity [16, 19, 39].

Since the first description of the concurrence between cutaneous fibrofolliculomas and colorectal cancer/ colon polyps [12], numerous studies have tried to prove this association in BHDS families with contradictory conclusions [3, 6, 7, 10, 13, 14, 28, 40, 41]. In our series, 5% of all carriers were affected by colorectal and/or colon dysplastic adenomas, but the size of the sample may not be adequate enough to draw a confident conclusion regarding this association. Moreover, a positive family history for these manifestations was noticed in 18% of BHDS families, yet this information should be interpreted with caution, since not all persons within each and every family underwent carrier testing for the familial disease causing variant. Whether colon cancer along with colonic polyps, which are quite frequent in the general population, may be a coincidental observation in BHDS or not, is still unclear and needs to be elucidated.

In conclusion, the current cohort provides valuable medical data accompanied by reports of eleven different pathogenic variants for a supposedly rare syndrome, among which one is an obviously common Swedish founder mutation. These data may allow a better understanding and evaluation of patients and families with a putative higher risk for life-threatening conditions such as spontaneous pneumothorax and renal cancer. The weakness in our study to assemble all necessary and essential clinicopathological features, for a more accurate BHDS assessment, is due to lack of national guidelines regarding diagnosis and management of the disorder. This problem may be similar in other Western countries as well. All scientific reports for BHDS though, illustrate the heterogeneity of this syndrome that often poses a diagnostic challenge to physicians. Therefore, a consensus that could provide up-to-date recommendations, regarding diagnostic and clinical management of BHDS is a necessary precondition in order to systematically ascertain the disorder.

## Supporting information

**S1 Table. Swedish cohort of BHDS.** Test result, variant and clinical phenotype is included for each individual. Each number represents one family and each letter represents one individual, ex 2a, 2b and 2c are three individuals which belong to the same family. N/A: Not available, CRC: Colorectal cancer, MDS: Myelodysplastic syndrome.
(XLSX)

**S2 Table. Family history of our cohort.** Pathogenic *FLCN* variant and clinical phenotype for each individual included in the study. FH: Family history, CRC: Colorectal cancer.
(XLSX)

## Acknowledgments

We would like to thank all patients and families who participated in this study.

## Author Contributions

**Conceptualization:** Kristina Lagerstedt-Robinson, Christos Aravidis.

**Data curation:** Kristina Lagerstedt-Robinson, Stefanos Tsiaprazis, Erik Björck, Emma Tham, Anna Poluha, Maritta Hellström Pigg, Ylva Paulsson-Karlsson, Magnus Nordenskjöld, Maria Johansson-Soller, Christos Aravidis.

**Formal analysis:** Kristina Lagerstedt-Robinson, Izabella Baranowska Körberg, Maria Johansson-Soller, Christos Aravidis.

**Methodology:** Kristina Lagerstedt-Robinson.

**Supervision:** Christos Aravidis.

**Writing – original draft:** Kristina Lagerstedt-Robinson, Izabella Baranowska Körberg, Maria Johansson-Soller, Christos Aravidis.

**Writing – review & editing:** Stefanos Tsiaprazis, Erik Björck, Emma Tham, Anna Poluha, Maritta Hellström Pigg, Ylva Paulsson-Karlsson, Magnus Nordenskjöld, Christos Aravidis.

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
