## [Decision Letter · Decision Letter 0]

10 May 2021

PONE-D-21-02346

A retrospective two centre study of Birt-Hogg-Dubé syndrome reveals a pathogenic founder mutation in FLCN in the Swedish population

PLOS ONE

Dear Dr. Aravidis,

Thank you for submitting your manuscript to PLOS ONE. After careful consideration, we feel that it has merit but does not fully meet PLOS ONE’s publication criteria as it currently stands. Therefore, we invite you to submit a revised version of the manuscript that addresses the points raised during the review process.

The reviewers found merit  in your manuscript but suggested revisions before it can be considered further. Please address those points with reasonable explanations and/or further analysis.

We look forward to receiving your revised manuscript.

Kind regards,

Obul Reddy Bandapalli, MSc, PhD

Academic Editor

PLOS ONE

Journal Requirements:

2) We note that you have included the phrase “data not shown” in your manuscript. Unfortunately, this does not meet our data sharing requirements. PLOS does not permit references to inaccessible data. We require that authors provide all relevant data within the paper, Supporting Information files, or in an acceptable, public repository. Please add a citation to support this phrase or upload the data that corresponds with these findings to a stable repository (such as Figshare or Dryad) and provide and URLs, DOIs, or accession numbers that may be used to access these data. Or, if the data are not a core part of the research being presented in your study, we ask that you remove the phrase that refers to these data.

Reviewers' comments:

Reviewer's Responses to Questions

**Comments to the Author**

1. Is the manuscript technically sound, and do the data support the conclusions?

Reviewer #1: Yes

Reviewer #2: Yes

2. Has the statistical analysis been performed appropriately and rigorously? 

Reviewer #1: N/A

Reviewer #2: Yes

3. Have the authors made all data underlying the findings in their manuscript fully available?

Reviewer #1: Yes

Reviewer #2: Yes

4. Is the manuscript presented in an intelligible fashion and written in standard English?

Reviewer #1: Yes

Reviewer #2: Yes

5. Review Comments to the Author

Reviewer #1: Aravidis and colleague investigated the incidence of BHDS caused by variants in FLCN gene in the Swedish population and describe its clinical manifestations. In addition, authors also found that Splice variant c.779+1G>T was the most common pathogenic variant, found in more than fifty percent of the Swedish population and thus suggesting a founder mutation in the Swedish population. This manuscript is well written and is an interesting read though I have concerned how this paper was structured.

1.Authors could add the sampling location: with a map of Sweden, locate geographic origins of families from where the samples were collected?

2.Authors could also determine the age of founder effect mutation in Swedish population, taking generation time as 30 years?

3. Since these mutations are present in the family, it is worth checking whether these are linked in family?

3.Talking about mutations, did authors also considered heterozygotes or only considered homozygotes in their analysis?

Minor comments:

4. Some sentences need to be checked again like “Single nucleotide variants were called using Mutation Identification Pipeline (MIP) (ref GitHub).” Please provide Github reference.

5. Figure resolutions are poor and should increase.

Reviewer #2: Gist/Summary: The authors come up analysing 278 sample cohort of a rare autosomal disorder, viz. Birt-Hogg-Dube syndrome (BHDS) and in the process identify a founder mutation besides pathogenic/likely pathogenic mutations.The work is largely linked to bridging the gap between the physician/clinicians and geneticists as the authors populate a metadata.

Strengths: A novel way of identifying both founder mutations and pat mutations.

The girth of the story is solid and very nicely written.

Weaknesses/Limitations: I feel the authors could have better exploited the work in classifying and matching the variants with rare disease registry which I feel they have subtly given a miss.

Minor but essential suggestions:

Did the authors find any mutations from non-coding and yet those coming from 5' UTRs/enhancers?

Any variants of unknown significance?

There must be a methodological flowchart which would be a very nice addition

The figures must be of high resolution

A work of such magnitude would be of great deal if this could be associated with a development of panel, for example n360

The authors could describe the applications of their study towards precision medicine

Scores on a scale of 0-5 with 5 being the best

Language: 4

Novelty:4

Scope/relevance: 4

Brevity: 3.75

6. PLOS authors have the option to publish the peer review history of their article (what does this mean?). If published, this will include your full peer review and any attached files.

Reviewer #1: No

Reviewer #2: **Yes: **Prashanth N Suravajhala

---

## [Author Response · Author response to Decision Letter 0]

11 Jun 2021

Response to reviewers has benn attached.

---

## [Decision Letter · Decision Letter 1]

27 Jul 2021

PONE-D-21-02346R1

A retrospective two centre study of Birt-Hogg-Dubé syndrome reveals a pathogenic founder mutation in FLCN in the Swedish population

PLOS ONE

Dear Dr. Aravidis,

Thank you for submitting your manuscript to PLOS ONE. After careful consideration, we feel that it has merit but does not fully meet PLOS ONE’s publication criteria as it currently stands. Therefore, we invite you to submit a revised version of the manuscript that addresses the points raised during the review process.

Though the revised manuscript is improved, one of the reviewer raised concerns that none of her comments were addressed. Please carefully address or give proper explanations to the points raised by the reviewers while resubmitting your revised manuscript.

We look forward to receiving your revised manuscript.

Kind regards,

Obul Reddy Bandapalli, MSc, PhD

Academic Editor

PLOS ONE

Reviewers' comments:

Reviewer's Responses to Questions

**Comments to the Author**

1. If the authors have adequately addressed your comments raised in a previous round of review and you feel that this manuscript is now acceptable for publication, you may indicate that here to bypass the “Comments to the Author” section, enter your conflict of interest statement in the “Confidential to Editor” section, and submit your "Accept" recommendation.

Reviewer #1: (No Response)

Reviewer #2: (No Response)

2. Is the manuscript technically sound, and do the data support the conclusions?

Reviewer #1: Partly

Reviewer #2: Yes

3. Has the statistical analysis been performed appropriately and rigorously? 

Reviewer #1: No

Reviewer #2: Yes

4. Have the authors made all data underlying the findings in their manuscript fully available?

Reviewer #1: Yes

Reviewer #2: Yes

5. Is the manuscript presented in an intelligible fashion and written in standard English?

Reviewer #1: No

Reviewer #2: Yes

6. Review Comments to the Author

Reviewer #1: Dear Authors,

None of the concerned raised were answered stating its out of the scope of the study and even the figures are not in high resolution.

Reviewer #2: Thank you for the responses. I am bit taken a back on some of the responses

The VUS makes sense in the adage of exomes and I am bit surprised why the authors did not look into them. " We only looked into pathogenic variants" perhaps may not augur well. Sorry!

The methodological flowchart would ensure a naive reader understand the wonderful work carried out by you. A PhD fellow of yours or a junior author can draw this. Methodological flowcharts carry an essence! Sincerely appreciate hat

7. PLOS authors have the option to publish the peer review history of their article (what does this mean?). If published, this will include your full peer review and any attached files.

Reviewer #1: No

Reviewer #2: **Yes: **Prashanth N Suravajhala

---

## [Author Response · Author response to Decision Letter 1]

27 Aug 2021

We provide an updated version as respond to reviewers as an attached file (Response to Reviewers_v2.docx).

---

## [Decision Letter · Decision Letter 2]

3 Feb 2022

A retrospective two centre study of Birt-Hogg-Dubé syndrome reveals a pathogenic founder mutation in FLCN in the Swedish population

PONE-D-21-02346R2

Dear Dr. Aravidis,

We’re pleased to inform you that your manuscript has been judged scientifically suitable for publication and will be formally accepted for publication once it meets all outstanding technical requirements.

Kind regards,

Obul Reddy Bandapalli, MSc, PhD

Academic Editor

PLOS ONE

Additional Editor Comments (optional):

Reviewers' comments:

Reviewer's Responses to Questions

**Comments to the Author**

1. If the authors have adequately addressed your comments raised in a previous round of review and you feel that this manuscript is now acceptable for publication, you may indicate that here to bypass the “Comments to the Author” section, enter your conflict of interest statement in the “Confidential to Editor” section, and submit your "Accept" recommendation.

Reviewer #1: (No Response)

Reviewer #2: All comments have been addressed

2. Is the manuscript technically sound, and do the data support the conclusions?

Reviewer #1: No

Reviewer #2: Yes

3. Has the statistical analysis been performed appropriately and rigorously? 

Reviewer #1: N/A

Reviewer #2: Yes

4. Have the authors made all data underlying the findings in their manuscript fully available?

Reviewer #1: Yes

Reviewer #2: Yes

5. Is the manuscript presented in an intelligible fashion and written in standard English?

Reviewer #1: No

Reviewer #2: Yes

6. Review Comments to the Author

Reviewer #1: (No Response)

Reviewer #2: I am convinced with all the changes rendered by the authors and thank them.

The tables and figures may be revisited before finalisng the proofs.

7. PLOS authors have the option to publish the peer review history of their article (what does this mean?). If published, this will include your full peer review and any attached files.

Reviewer #1: No

Reviewer #2: **Yes: **Prashanth Suravajhala

---

## [Editor Report · Acceptance letter]

7 Feb 2022

PONE-D-21-02346R2 

A retrospective two centre study of Birt-Hogg-Dubé syndrome reveals a pathogenic founder mutation in *FLCN* in the Swedish population 

Dear Dr. Aravidis:

I'm pleased to inform you that your manuscript has been deemed suitable for publication in PLOS ONE. Congratulations! Your manuscript is now with our production department. 

Kind regards, 

on behalf of

Dr. Obul Reddy Bandapalli 

Academic Editor

PLOS ONE